# Disruption of Endochondral Ossification and Extracellular Matrix Maturation in an Ex Vivo Rat Femur Organotypic Slice Model Due to Growth Plate Injury

**DOI:** 10.3390/cells12131687

**Published:** 2023-06-22

**Authors:** Vanessa Etschmaier, Muammer Üçal, Birgit Lohberger, Markus Absenger-Novak, Dagmar Kolb, Annelie Weinberg, Ute Schäfer

**Affiliations:** 1Research Unit for Experimental Neurotraumatology, Medical University of Graz, 8036 Graz, Austria; vanessa.mair@medunigraz.at (V.E.); muammer.uecal@medunigraz.at (M.Ü.); 2Department of Orthopaedics and Trauma, Medical University Graz, 8036 Graz, Austria; birgit.lohberger@medunigraz.at (B.L.); anneliemartina.weinberg@medunigraz.at (A.W.); 3Bio-Tech-Med Graz, 8010 Graz, Austria; 4Center for Medical Research, Core Facility Imaging, Medical University of Graz, 8036 Graz, Austria; markus.absenger@medunigraz.at; 5Center for Medical Research, Core Facility Ultrastructure Analysis, Gottfried Schatz Research Center, Medical University of Graz, 8010 Graz, Austria; dagmar.kolb@medunigraz.at; 6Division of Cell Biology, Histology and Embryology, Gottfried Schatz Research Center, Medical University of Graz, 8010 Graz, Austria

**Keywords:** bone metabolism, injury repair, organotypic culture, endochondral ossification

## Abstract

Postnatal bone fractures of the growth plate (GP) are often associated with regenerative complications such as growth impairment. In order to understand the underlying processes of trauma-associated growth impairment within postnatal bone, an ex vivo rat femur slice model was developed. To achieve this, a 2 mm horizontal cut was made through the GP of rat femur prior to the organotypic culture being cultivated for 15 days in vitro. Histological analysis showed disrupted endochondral ossification, including disordered architecture, increased chondrocyte metabolic activity, and a loss of hypertrophic zone throughout the distal femur. Furthermore, altered expression patterns of Col2α1, Acan, and ColX, and increased chondrocyte metabolic activity in the TZ and MZ at day 7 and day 15 postinjury were observed. STEM revealed the presence of stem cells, fibroblasts, and chondrocytes within the injury site at day 7. In summary, the findings of this study suggest that the ex vivo organotypic GP injury model could be a valuable tool for investigating the underlying mechanisms of GP regeneration post-trauma, as well as other tissue engineering and disease studies.

## 1. Introduction

Postnatal longitudinal bone growth primarily takes place within the growth plate (GP) of long bones under a tightly orchestrated process termed endochondral ossification. Here, the cartilage template is replaced by bone in a continuous process until adulthood is reached. Cartilage is the weakest part of growing bone and in 15–30% of all bone fractures, the GP is affected by the sustained injuries [1,2,3]. According to the Salter–Harris classification system (Type III to VI) the associated complications as a consequence of such injuries include delayed union or regeneration through bone formation, which can lead to growth discrepancies in length and form, reducing patient quality of life in the long-term [4,5,6].

At present, the underlying molecular and cellular processes of GP trauma remain unclear and sufficient therapy strategies do not exist. Studying bone repair at both the cellular and molecular levels has proved difficult. For instance, translating the findings obtained from conventional two-dimensional (2D) cell culture in vitro to an in vivo situation is often challenging [7], since the complex architecture of the bone, the mineralization process (which is particularly important in bone maturation), and the interaction of different cell-type activities cannot be mirrored—something that has long been known to be essential for ex vivo bone formation [8]. Therefore, animal studies are often implemented to better understand the pathophysiological processes of bone growth impairment. However, in addition to the ethical implications, such animal models are expensive and require a large number of personnel, much labor, and a large sample size [9,10,11]. To reduce animal experiments and to overcome the discrepancy between these and conventional 2D cell cultures, 3D organotypic cultures (OTCs) were developed. These models were produced to achieve an in vivo-like 3D structure with characteristic biological function. They mimic in vivo conditions in terms of specific cell behavior, morphological transitions, cell maturation and differentiation, and migration within the tissue microenvironment [12]. Here, Srinivasaiah et al. overcame the aforementioned limitations of 2D cell cultures by cultivating postnatal femoral bone slices ex vivo. Our group demonstrated endochondral ossification within the distal femur over a 15-day cultivation period. Using histology, immunochemistry, and ATR-IR spectrometry, highly structured chondrocyte maturation was observed including the time-dependent increase and subsequent disintegration of hypertrophic chondrocytes that was paralleled by a mineralization process. These findings were supported by an increase in chondrogenic and osteogenic markers at the respective time points evaluated [13].

We hypothesized that GP injury in this ex vivo organotypic femur model will lead to pronounced alterations in the highly structured endochondral osteogenesis processes. Therefore, we evolved the established ex vivo organotypic bone model [13] and applied a GP injury that allows the real-time investigation of underlying cellular and pathophysiological regeneration processes following trauma. This model allowed the visualization of tissue architecture and changes in cell population e.g., chondrocytes, mesenchymal stem cells (MSC), osteoblasts, and chondroclasts during physiological and pathophysiological growth plate regeneration.

Four-day-old postnatal rat femurs were cut into 300 µm thick slices and cultured in osteogenic medium [13]. Subsequently, a 2 mm long horizontal cut was introduced from the later secondary ossification center through the GP and into the bone marrow cavity. Tissue viability and cellular changes and alterations in extracellular matrix (ECM) composition were analyzed ex vivo at the day of injury (0) and at days 1, 3, and 7, and by live/dead staining, histology, immunohistochemistry, and electron microscopy.

## 2. Materials and Methods

### 2.1. Preparation and Culturing of Ex Vivo Organotypic Bone Slice Culture (OTCs)

Animal handling and organ collection were performed in accordance with the national laws on animal experimentation in Austria (TVG 2012). Femurs from 4-day-old Sprague Dawley rats (12.0 g +/−2.1 g) were quickly explanted after decapitation and washed in ice-cold Dulbecco’s phosphate buffer saline (DBPS) (Thermo Fisher Scientific, Invitrogen, Waltham, MA, USA). Subsequently, the femurs were embedded in molten agar cast onto prechilled stainless steel molds (Sakura, Tokyo, Japan) and placed on ice for fast polymerization into an agar block. The agar block was mounted onto the specimen plate using cyanoacrylate tissue culture grade glue (Roti^®^ coll 1; Carl Roth, Karlsruhe, Germany). After calibration of a VT1000 Leica vibratome according to the manufacturer’s instructions, 300 µm coronal sections were obtained under buffered conditions using ice-cold 1x DPBS and a stainless steel razor blade (Gillette Procter & Gamble, Cincinnati, OH, USA) at an amplitude of 1.5 m and sectioning speed of 0.06 m/s using a consistent blade angle. With a 2 mm long razor blade, an incision from the later secondary ossification center along the axes of the bone through the GP and metaphysis into the bone marrow cavity was made to imitate an injury through the GP within the left femur. The respective right femur from the same animal served as a control and was cultivated without injury. The slices were then gently washed in minimum essential medium Eagle (MEM) (Sigma-Aldrich, St. Louis, MO, USA) + penicillin (100 U/mL) + streptomycin (100 µg/mL) (Invitrogen) for 30 s and transferred into 6-well plates containing a Millcell cell culture 30 mm insert with a pore size of 0.4 µm (Merck, Darmstadt, Germany). Each well was supplemented with 1 mL of osteogenic medium containing MEM with 5% horse serum (Invitrogen), 1% penicillin/streptomycin, 1X Insulin-Transferrin-Selenium, 100 μm ascorbic acid, 10 mM β-glycerophosphate (both Sigma-Aldrich), 10 nM dexamethasone (STEMCELL Technologies, Vancouver, Canada), and 0.01 µg/mL TGFβ-3 (Thermo Fisher Scientific). Every day, 500 µL of medium was replaced with fresh medium. The 6-well plates were incubated in a humidified atmosphere at 37 °C with 5% CO_2_. Slices were analyzed to investigate distinct phases of the regeneration processes, namely the inflammatory phase (days 1–3), the fibrogenic phase (days 3–7), and the osteogenic and maturation phase (days 8+) [14]. To this end, slices were harvested on the day of preparation (0) and days 1, 3, 7, and 15 of in vitro cultivation (see Figure 1a).

### 2.2. Tissue Processing for Histological Evaluation and Analyses

Cultured slices were fixed in 2 mL 4% paraformaldehyde (PFA; MERCK KGaA, Darmstadt, Germany) overnight at +4 °C. Slices were processed without decalcification using a Sakura Tissue Processor VIP 5E-F2 processor (Sakura, Tokyo, Japan) and subsequently embedded in paraffin. Five-micrometer serial sections were obtained using a Microtome HM 360 rotation microtome (Hyland Scientific GmbH, Berlin, Germany). Sections were incubated overnight at +60 °C, deparaffinized by two 10 min washes in Roticlear (CarlRoth, Karlsruhe, Germany), and then rehydrated in a series of descending alcohol concentrations (100% for 5 min and 95% and 70% for 2 min each) and a brief wash in distilled water. After staining, whole object slides were scanned with Aperio ScanScope (Leicabiosystems, Vienna, Austria). Afterward, images were generated with Image Scope software (v12.4.0.50.43, Leicabiosystems). Quantitative analyses were performed with ImageJ (v1.53f51) (U. S. National Institutes of Health, Bethesda, MD, USA). To assess the morphological differences/changes between the injured bone and their respective controls, the postnatal femur was empirically divided into distinctive zones—transition zone (TZ), middle zone (MZ), and deep zone (DZ)—based on the maturation and differentiation stage of chondrocytes and their cell arrangement [13].

### 2.3. Live and Dead Staining

Cultured bone slices were washed in 1x DPBS and incubated in 2 µM calcein acetoxymethyl (AM) and 4 µM ethidium homodimer -1 (EthD-1) (Gibco, Invitrogen) working solution, prepared in 1x DPBS (Gibco) containing calcium and magnesium for 40 min. Tissue slices kept in 70% ethanol for 1 h represented dead tissue, and, therefore, served as negative controls. After incubation in the working solution, tissues were washed 3 times in 1x DPBS and transferred to a 6-well plate. Laser Confocal imaging (Zeiss SM 510) (Carl Zeiss Microscopy GmbH, Jena, Germany) was performed under 10X objective at an excitation wavelength of 488 nm and 543 nm with an open pinhole. The slices were examined at days 0, 1, 3, 7, and 15 of in vitro cultivation. The images were analyzed using ZM 2009 software (Zen 2009; Carl Zeiss Microscopy GmbH).

### 2.4. Electron Microscopy

Bone slices were fixed in 2.5% (wt/vol) glutaraldehyde and 2% (wt/vol) PFA in 0.1 M cacodylate buffer, pH 7.4, for 2 h, postfixed in 1% (wt/vol) osmium tetroxide for 2 h at room temperature (RT). After dehydration in a graded series of alcohol, tissue was infiltrated with ethanol and TAAB epoxy resin (pure TAAT epoxy resin) and replaced in TAAB epoxy resin (8 h), transferred into embedding molds, and polymerized for 48 h at 60 °C. Ultrathin bone sections (70 nm) were cut with a UC 7 Ultramicrotome (Leica Microsystems, Vienna, Austria) and stained with lead citrate for 5 min and platin blue for 15 min. Electron micrographs were taken using a Tecnai G2 transmission electron microscope (FEI, Eindhoven, Netherlands) with a Gatan UltraScan 1000 charge-coupled device (CCD) camera (−20 °C; acquisition software Digital Micrograph; Gatan, Germany). The acceleration voltage was 120 kV. Cells were identified based on their distinct morphological appearance, namely, their nucleus and shape, which was compared to cells gained from the cell culture or control slices.

### 2.5. Histological Stainings

All histological and immunohistochemistry stainings were performed with three biological replicates, each represented by three sections obtained at different depths along the dorsoventral axis.

### 2.6. Hematoxylin and Eosin Staining

Rehydrated tissue sections were soaked in Mayer’s hematoxylin (Gatt-Koller, Absam, Austria) for 3 min, followed by a 10 min rinse under warm, running tap water. A quick wash in distilled water and 95% ethanol was followed by eosin staining (Gatt-Koller) for 1 min. Stained sections were dehydrated by an increasing series of alcohol concentrations, which were finished by a double wash in Roticlear. Finally, sections were mounted with a non-aqueous mounting media (Rotimount, CarlRoth).

### 2.7. Movat’s Pentachrome Staining

Movat’s pentachrome staining enables specific staining and differential visualization of various tissue components with high differentiation of various hard and soft tissue components. Deparaffinized and rehydrated bone sections were stained with 0.1% Alcian blue solution (CarlRoth) and stabilized in 10% alkaline ethanol (CarlRoth), followed by staining with Weigert’s iron hematoxylin (A+B 1:1, CarlRoth), brilliant crocein (Sigma-Aldrich) acid fuchsin (CarlRoth) solution. Differentiation with phosphotungistic acid (CarlRoth) and lastly, staining with Gatinais saffron staining (Chroma-Waldeck, Münster, Germany) concluded the protocol. Sufficient washing steps under running tap water and distilled water cleaning between staining steps completed the procedures. The different structures resulted in cell nuclei (black), cytoplasm (reddish), elastic fibers (red), mineralized tissue/bone (bright yellow), and cartilage (green).

### 2.8. Safranin O/Fast Green Staining

Safranin O binds to glycosaminoglycan and stains areas representing proteogolycans in the cartilage [15]. Sections were stained in Weigert’s iron hematoxylin solution followed by a wash under running tap water. Staining with 0.1% Fast Green (Sigma-Aldrich) and subsequent differentiation in 1% acetic acid solution and a final staining with 0.1% Safranin O (CarlRoth) solution finished the procedure.

### 2.9. Immunohistochemistry

Extracellular matrix proteins (markers of chondrogenesis) were assessed with immunohistochemistry and the Ultra Vision LP/HRP rabbit/mouse kit (Epredia, Kalamazoo, MI, USA). After deparaffinization and rehydration, sections were processed for antigen retrieval either in a decloaking chamber (95 °C, 10 min) with 1 mM EDTA (pH 8.0) or in a water bath (60 °C overnight) with 10 mM Sodium citrate buffer (pH 6.0, with 0.05% Tween-20). Endogenous peroxidase activity was blocked with 0.3% hydrogen peroxide (MERCK, Darmstadt, Germany) at room temperature. Non-specific protein binding was blocked with the respective protein block from the kit. Sections were then incubated with the primary antibody overnight at 4 °C. On the primary antibodies of mouse origin, a bridging step with the primary antibody enhancer from the kit was performed before all sections were treated with HRP-enzyme followed by development of the positive staining with AEC (Abcam, Cambridge, UK). Hematoxylin was used for counterstaining. The optimized steps of the immunohistochemistry protocol for each antibody are given in Table 1.

### 2.10. Data Presentation and Statistical Analyses

Results are presented as the mean  ±  SEM of the independent variables. Quantification was obtained from a minimum of three experimental replicates and three sections per bone slice at different depths. Individual data points were superimposed on the histograms to provide additional information on the data distribution. Statistical analyses were performed using IBM SPSS Statistic 26 (version 6, New York, NY, USA) and GraphPad InStat software (GraphPad Prism 9.3.1 (471), Boston, MA, USA). Data were tested for normality with the Kolmogorov–Smirnov test. Since the data distribution in all samples significantly deviated from a normal distribution, the statistical significance of the observed differences was tested with non-parametric tests. Single comparisons were tested using the Mann–Whitney U test. Multiple comparisons were tested with the Kruskal–Wallis H test followed by pairwise analysis with Bonferroni correction. A difference with *p* ≤ 0.05 was deemed statistically significant in all statistical assessments.

## 3. Results

### 3.1. Experimental Setup

Four-day-old postnatal rat femurs were processed into 300 µm ex vivo bone slice cultures and a GP injury was inflicted as shown in the schematic summary of Figure 1a. To assess the morphological changes and differences between injured bone and their respective controls, changes in the maturation and differentiation stages of chondrocytes and their cell arrangement are described for distinct epiphyseal zones, i.e., the transition zone (TZ), the middle zone (MZ), and the deep zone (DZ) [12,13] (Figure 1b).

To examine the underlying processes during endochondral ossification and its repair mechanisms, the injured bone slices and the corresponding controls were cultured for 0, 1, 3, 7, and 15 days and analyzed histologically with hematoxylin and eosin (H&E), Movat’s Pentachrome, and Safranin O/Fast Green stains (Appendix A). Pronounced differences were observed starting at day 7; therefore the results at day 7 and day 15 in the ex vivo culture were described in more detail below. As a reference for GP injury repair, Figure 1c shows the maximum GP injury area inflicted on the day of preparation (0).

### 3.2. Epiphyseal Tissue Architecture Changes Due to Growth Plate Injury

In H&E staining, a structured organization of the epiphysis and GP was observed at day 7 in non-injured controls (Figure 2i). A high number of single round-shaped chondrocytes with large nuclei (black arrows, black box), as well as prehypertrophic chondrocytes located adjacent to the potential secondary ossification center (red arrows, red box), were observed in the resting zone of the TZ. The MZ was predominantly occupied by stacks of proliferating isogenic groups of chondrocytes (blue arrows, blue box) and elongated prehypertrophic chondrocytes (orange arrows, orange box). Large lacunae with single cells indicating chondrocyte hypertrophy (purple arrow) were observed in the DZ adjacent to enlarged empty lacunae, indicating the disintegration of hypertrophic chondrocytes (green arrow) in the GP. The structured organization of the chondrocyte transition throughout the epiphysis is characteristic of the highly regulated process of chondral ossification and is in accordance with our previously published findings reporting the ex vivo culturing of postnatal femur slices [13]. This organization of chondrocyte transition was significantly impaired in GP-injured bone slices at day 7.

Within the TZ of the injured bone slices at day 7, chondrocytes and prehypertrophic chondrocytes were randomly distributed (Figure 2ii; black and red arrows). In the MZ, a condensed area of stacked proliferating chondrocytes was observed close to the GP (black dashed circle). Similarly, the hypertrophic and calcification area in the DZ was profoundly decreased to only a couple of layers of hypertrophic chondrocytes (purple rectangle). However, the dark staining of the matrix septa might indicate early mineralization taking place within this area. Furthermore, cell aggregates within the injury cleft of the DZ were observed (red dashed oval).

At day 15, randomly distributed chondrocytes and prehypertrophic chondrocytes were also observed in the TZ and MZ of the control slices (Figure 2iii; black and red arrows). The number and area of stacked proliferating chondrocytes were reduced in the MZ and the initiation of a mineralizing process in the DZ as indicated by the intense dark staining along the matrix septa was observed (black rectangle). At day 15, chondrocyte transition had also advanced in the injured slice model (Figure 2iv). In the TZ, chondrocyte (black arrow) transition into prehypertrophic chondrocytes (red arrows) adjacent to the potential secondary ossification center was observed with more organization. However, in the MZ, a dense layer of chondrocytes (black arrows) was sandwiched between prehypertrophic chondrocytes adjacent to the potential secondary ossification center and clusters of multicellular columns towards the DZ (black oval). In some of the injured slices, the injury cleft was closed at the hypertrophic area (red dashed oval). The time-dependent chondrocytes were associated with changes in H&E staining, which was confirmed by the epiphyseal overview summarizing day 0 to day 15 for both the control and injured slices (Appendix A).

### 3.3. Injury-Associated Augmented Cartilage Generation and Attenuated Ossification Process

To explore the effect of GP injury on cartilage maturation and endochondral ossification, Movat’s Pentachrome staining was performed. At day 7, in control slices, profound cartilage-stained areas were mainly observed in the TZ adjacent to the prospective articular cartilage and in the MZ adjacent to the zone of Ranvier (Figure 3i, black stars). Cartilage staining of the interterritorial matrix along the DZ was marginal. However, in injured slices, profound cartilage staining was detected throughout the entire TZ and adjacent to the hypertrophic area in the MZ and DZ (Figure 3ii, black stars) at this time point. Cell aggregates within the cleft area that were also observed within the H&E staining were detected as intense red/brownish stained cells in Movat’s Pentachrome staining (Figure 3ii, DZ, black dashed oval). At day 15, the distal femurs exhibited clear differences between injured and control bone slices. Cartilage was limited to the emerging articular cartilage area (Figure 3iii, overview dark green staining) and the area adjacent to the zone of Ranvier (MZ, black stars) in the control slices. In contrast, in the injured bone slices, a wide “ring” of intensely stained cartilage encases the epiphysis (Figure 3iv, overview dark green staining). The pronounced staining indicates an intensified secretion/assembly of cartilage proteins in this area.

The control and injury-specific differences in the time-dependent and regional secretion of cartilage proteins are underlined by Safranin O/Fast Green staining of proteoglycan (PG). The intensity of the staining depends on the amount of PG present within the tissue. In control slices, PG was observed in the periphery of the epiphysis at day 7 (Figure 4i). In the injured slices, intensive PG staining was also detected in the MZ at this time point (Figure 4ii). This is in accordance with the presence of proliferating chondrocytes in the MZ of the injured bone at this time point versus prehypertrophic chondrocytes in the control (compare Figure 2). At day 15, intensive PG staining was confined to the emerging articular cartilage in the control tissue and the periphery of the epiphysis in injury slices (Figure 4iii,iv, overview dark red staining). Furthermore, a high number of intensely stained cell membranes were also observed in the epiphysis periphery of injured slices at this time point. This indicates that PG is newly synthesized and transported to the cell membrane/ECM as part of the regeneration process due to GP injury (black arrows, black boxes). Specific PG staining patterns in the time-dependent injury and control slices match the Movat’s Pentachrome staining patterns.

Our histological staining results indicate that injury induced the disruption of the highly structured endochondral ossification process towards an augmented cartilage generation and an attenuated osteoid formation. This is implied by the seemingly disorganized chondrocyte transition and the reduced hypertrophic zone. These findings were further substantiated by the immunohistochemical analysis of type II collagen (Col2α1), the major fibril-forming collagen in cartilage, aggrecan (Acan), a major proteoglycan found in articular cartilage, and collagen type X (ColX), a marker for hypertrophy. The time-dependent expression pattern and intensity of Col2α1 and Acan matched the region-specific changes in cartilage generation in injured and control slices (for comparison see Appendix A; immunohistochemical controls are depicted in Appendix A). Furthermore, in accordance with the reduced hypertrophic zone in the injured slices, time-dependent ColX expression was also diminished in such slices at all time points (for comparison see Figure 1 and Appendix A).

### 3.4. Stem Cell Infiltration at the Injury Site

Although we observed a pronounced disruption of endochondral processes in the injured slices, there was also evidence of the initiation of regenerative processes. In H&E, Movat’s Pentachrome, and Safranin O/Fast Green stainings, a cell cluster was detected close to the GP within the injury site at day 7. At day 15, the injury gap was closed in this region. In order to substantiate the manifestation of a regenerative process, we attempted to detect cells within the injury site at different time points by magnifying these regions in H&E and Movat’s Pentachrome staining, as well as in Col2α1 expression analysis.

H&E and Movat’s Pentachrome staining revealed injured tissue at day 0 with regeneration processes beginning at day 1 (Figure 5a,b). Starting at day 1, we observed a number of cells that seemed to span the injury cleft, and the number of cells in this area increased over the 15-day cultivation period. Furthermore, we observed fibers and tissue that seemed to originate from dissolving chondrocytes directly beside the injury. We further noted that the nuclei of the cells in the immediate vicinity of the injury were oriented towards the injury cleft (Figure 5a, day 3 and Figure 5b, day 1). We also detected cell aggregates at the hypertrophic zone at day 7 (Figure 5b, black dashed oval) and in the upper part of the injury at day 15. The location of these cell aggregates indicates a possible migration of bone marrow cells towards and along the injury cleft. Col2α1 IHC staining revealed positive staining within the injury site at day 0 but not in the rest of the epiphysis (Figure 5c). At day 1, the edge of the injury cleft was intensely Col2α1-stained, indicating the onset of regeneration processes due to increased Col2α1 production by surrounding chondrocytes. Fibers observed within the injury cleft were also positively stained for Col2α1 at day 3, as well as in the rest of the epiphysis (see also Appendix A). Cell aggregates observed in H&E and Movat’s Pentachrome staining at day 7 were embedded in an intensely stained Col2α1 matrix (Figure 5c, black dashed oval). By day 15, the majority of the injury was filled with small cells embedded in a Col2α1-positive matrix (Figure 5c, 15 DIV, black box).

In order to verify the cellular status seen within histology, we performed live/dead staining in bone slices at 0, (1, 3,) 7, and 15 days in vitro. Here, dead cells and/or tissue were observed within the injury site at day 0, which disappeared by day 7 (Figure 6a, red cells). At day 7, elongated live cells both surrounded and lined the edges of the injury site and increased in number to completely cover the injury site at day 15 (Figure 6a, white arrows). Furthermore, the dead cell count within the GP injury area revealed a constant significant increase in dead cells over the cultivation time in both injured 0–7 (z = −2.85, *p* = 0.044) and 0–15 (z = −3.64, *p* = 0.003) and non-injured control 0–7 (z = −3.19, *p* = 0.01) and 0–15 (z = −3.49, *p* = 0.01) bone slices (Figure 6c,d). However, no statistically significant differences were observed between the injury and the control at the respective time points (Figure 6b). Respective controls are presented in the Appendix A.

The injury cleft and the cells involved in the repair processes were visualized using scanning transmission electron microscopy (STEM) with field emission mode in combination with ATLAS TM at day 0 and day 7. This combined method enabled imaging of a greater area of the injury cleft, which was then recorded by the transmission electron microscope. At day 0, cell fragments (purple stars) and erythrocytes (yellow arrow) were detected within the injury site (Figure 7a,b). By day 7, cell clusters containing stem cells (black arrowheads), fibroblasts (black arrow), and chondrocytes (white arrowhead), producing collagen fibers (black star) within the injury site were observed (Figure 7c,d). We noted that the cluster located at the trabecular spicules moved upward along the DZ within the injury cleft. Together with the cell aggregates seen within the H&E or Movat’s Pentachrome staining, this indicated that the regeneration of GP injury progresses from the trabecular spicules upwards. Observed cells were identified based on their characteristic ultrastructural morphology in collected stem cell pellets and were validated against chondrocytes and fibroblasts from control bone slices (Figure 7e–g).

## 4. Discussion

The regeneration of a GP injury back to its original cartilaginous tissue form is crucial for undisrupted postnatal bone development. Using an ex vivo organotypic femur culture, an injury through the GP into the bone marrow cavity, comparable to the Salter–Harris type IV classification, was applied to investigate the influence of GP injury on postnatal endochondral osteogenic development, as well as to study the underlying molecular and cellular GP injury repair mechanisms.

Recent studies suggest that the growth discrepancies after GP injury are the result of a decreased rate of proliferation and concomitant alteration of the structural architecture of the GP [16,17,18]. We observed a disruption of GP architecture and organization due to GP injury at the local injury area and the distant areas of the epiphysis (H&E, Movat’s Pentachrome, and Safranin O/Fast Green staining). Most significantly, a loss of hypertrophic zone and a random distribution of chondrocyte maturation stages within the DZ were noted due to GP injury at the later time points. These results are in line with other studies, which conclude that GP injury is associated with bone development disturbances, as well as growth arrest due to a decrease in GP thickness [19].

To evaluate tissue maturation, we performed Movat’s Pentachrome staining to examine the state of endochondral ossification after GP injury with respect to in vitro regeneration. Here, our results indicated that the endochondral ossification is disrupted by day 7 postinjury throughout the whole distal epiphysis due to the GP injury. It seems that (hyaline) cartilage, which is usually limited to the articular cartilage area, is distributed in a “ring”-like formation underneath the perichondrium. Furthermore, premature mineralization within the TZ due to sustained GP injury was observed. Articular cartilage is subject to severe biomechanical forces and its function is to withstand these compressions and to transport the forces with a low coefficient of friction to prevent injury [20]. Interestingly, premature mineralization of the ECM, which we also observed as a consequence of the GP injury within the TZ, together with increases in early osteogenic differentiation of mesenchymal and preosteoblast cells, is thought to lead to abnormal skeletal development [21].

Since PGs are a dominant component of cartilage, and tissue maturation was disrupted upon Movat’s Pentachrome staining, Safranin O/Fast Green staining was performed to determine PG distribution. An increased number of chondrocytes with profound positive PG staining of the capsular and territorial matrix of the injured bone was found when compared to the control. Chondrocytes are able to change their metabolic activity rapidly in response to mechanical stress and/or sustained injury, ensuring the durability of cartilage under strong biomechanical forces and its recovery after injury. However, in case of extensive injury, repair mechanisms most likely fail and result in degenerative processes e.g., osteoarthritis (OA) [22]. Notably, Siffert et al. described a rise in chondrocyte metabolic activity within the articular and growth cartilage for self-repair after a sustained injury. It was further stated that this metabolic increase is short-lived and would lead to degenerative changes within the matrix [23].

Regarding the indication of an altered cartilage ECM and observed increases in the number of highly metabolic active chondrocytes due to GP injury, the other main components of the ECM and markers of chondrocyte maturation, Col2α1, ColX, and Acan were inspected. An increase in chondrocyte metabolic activity with respect to matrix secretion was detected, as well as impaired chondrocyte transition into the hypertrophy state starting at day 3 in vitro, with a peak at day 7 in vitro due to the sustained GP injury. In particular, chondrocytes within the TZ and DZ seemed to alter their maturation state by changing their ECM expression pattern of Col2α1, ColX, and Acan postinjury compared with the non-injured bone slices.

Col2α1, the major collagen fibril of the cartilage ECM, is mainly released by committed and proliferating chondrocytes and within the ECM, marks the transition from cartilage to bone [24,25]. As part of the cartilage ECM, Col2α1, among others, forms an interconnected network with the main PG Acan. Together, they are capable of binding to hyaluronic acid, growth factors, and other molecules, enabling the cartilage tissue to withstand mechanical forces, maintain physiological homeostasis, and act as regulating factors similar to soluble signals [26]. Chondrocytes increase their cell volume by ~10-fold before they become apoptotic facilitating longitudinal bone growth [27]. With this process, terminal hypertrophic chondrocytes secrete ColX into the ECM, thus promoting calcification [28].

Previous studies demonstrated that bone bridge formation is accompanied by a lack of Col2α1 and ColX expression within the GP injury area [29,30]. In accordance with these studies, we observed a decrease in Col2α1 secretion and a profound decrease in hypertrophy marker ColX after GP injury. Interestingly, this process is also seen within degenerative articular cartilage [31].

Bone fracture regeneration undergoes four stages to restore the original tissue structure: (1) initial rapid injury-induced inflammatory response with the formation of a hematoma, (2) primary intramembranous bone formation, (3) chondrogenesis, and (4) endochondral ossification and remodeling [14,32].

Although bone bridge formation after GP injury is common, we did not observe any bony regeneration in our model. However, our bone slices exhibited both altered ECM composition and chondrocyte metabolic activity. Even though bone bridge formation after a sustained GP injury has long been assumed to be the main reason for bone length discrepancies, recent studies have demonstrated that the formation of a bone bridge may only play a minor role and might only develop when certain areas of the GP are affected [33,34,35]. For instance, Wong et al. [36] showed reduced limb length discrepancy despite bone bridge formation when they treated a growth plate injury with mesenchymal stem cell (MSC) exosomes [37].

Despite the reliability and robustness of this methodology, the lack of vascularization and systematic influences due to the absence of blood flow must be considered when interpreting the results. Nevertheless, it has been suggested that these factors are not critical for short-term studies of bone development [38]. A further limitation of our system is the absence of immune cells. Particularly in the first phases of injury repair (inflammatory phase), immune cells are the first to respond upon injury and help to eliminate debris and secrete stimulating molecules to recruit stem cells and progenitors to the site of injury. Therefore, the effects of the immune cells are not short-lived but long-lasting and contribute to the restoration of normal bone growth and development [39,40]. Chung et al. [41] found that without the inflammatory phase, mesenchymal stem cell infiltration into the injury site was not delayed. However, by day 10 postinjury, the osteogenic transcription factor cbf-alpha1 and protein osteocalcin were upregulated while chondrogenic transcription factor Sox9 and ECM protein Col2α1 were downregulated, suggesting bony repair rather than cartilaginous regeneration. This is in line with our SEM imaging results, which reveal the presence of stem cells, fibroblasts, and chondrocytes within the injury site at day 7 postinjury. Furthermore, live/dead staining showed the formation of a cellular network starting at day 7 in vitro, indicating a migration of cells from the zone of Ranvier towards the injury site. These findings further support the results from other studies, which showed that the zone of Ranvier is not only a stem cell niche but also contains different populations of progenitor cells with the ability to migrate toward the injury site [42,43]. Shirtomoto et al. [44] demonstrated the ability of mechanically damaged cartilage chondrocytes, especially from the superficial zone, to migrate towards the injury area and bridge the gap fully, rather than a random migration.

## 5. Conclusions

The identification of pathological GP repair mechanisms and the underlying processes in altered chondrocyte metabolism and differentiation is essential in preventing or treating bone length discrepancies due to GP injury. The data obtained using the ex vivo organotypic femur culture model represent an important contribution to the understanding of these pathological processes. This methodology allows for exceptional accessibility without the need for harsh treatments such as decalcification, making this a significant methodology to study bone regeneration/development and bone diseases or assist with screening for suitable biomaterials and pharmaceutical agents.

## Figures and Tables

**Figure 1 cells-12-01687-f001:**
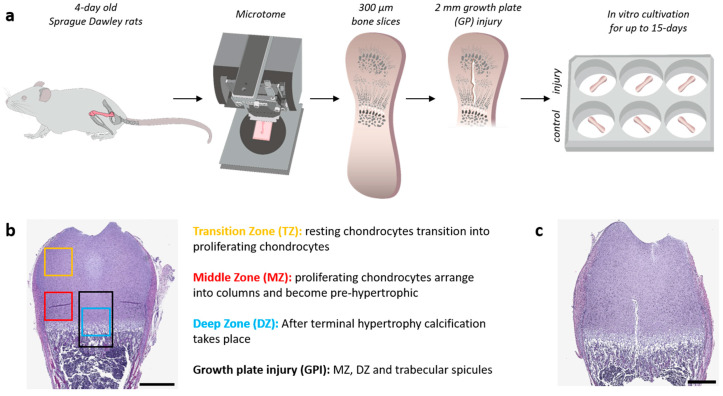
Experimental setup of ex vivo model preparation and zone specification. (**a**) 4-day-old Sprague Dawley rats were sacrificed and the left and right femur were explanted and processed into 300 µm thick bone slices by vibratome. With a 2 mm long razor blade, a vertical injury from the center of the epiphysis through the growth plate (GP) into the bone marrow cavity was inflicted in the left femur slice; (**b**) The respective right femur slices served as a control. Bone slices with and without injury were cultivated for up to 15 days. (**b**) Hematoxylin and eosin staining (H&E) shows the specific zone evaluated during histological staining. Transition Zone (TZ, yellow), Middle Zone (MZ, red), Deep Zone (DZ, blue), and growth plate injury area (GPI, black). (**c**) Maximum injury inflicted on the day of preparation (0). Scale bar = 500 µm.

**Figure 2 cells-12-01687-f002:**
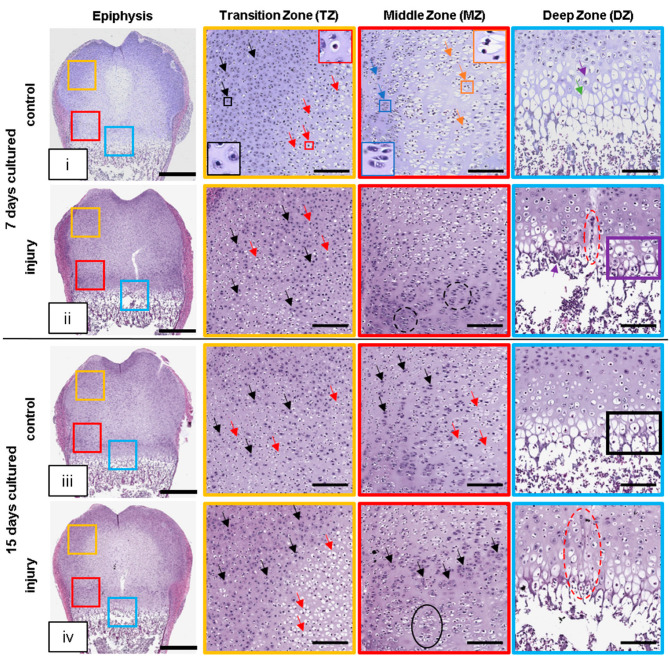
Hematoxylin and eosin (H&E) staining demonstrated the interruption of endochondral ossification due to GP injury. (**i**–**iv**) Representative images of rat distal femur OTCs with and without injury at day 7 and day 15 and their respective controls stained with H&E. Colored squares represent the transition zone (TZ, yellow), middle zone (MZ, red), and deep zone (DZ, blue). Black arrow = active chondrocytes (black box with higher magnification), red arrow = prehypertrophic chondrocytes (red box with higher magnification), blue arrow = proliferating chondrocytes (blue box with higher magnification), orange arrow = (pre)hypertrophic chondrocytes (orange box with higher magnification), purple arrow = chondrocyte hypertrophy, green arrow = disintegrated hypertrophic chondrocyte, black dashed circle = stacks of proliferating chondrocytes, purple rectangle = layers of hypertrophic chondrocytes, black rectangle = matrix septa, and black oval = clusters of multicellular columns; scale bar: epiphysis = 500 µm, TZ/MZ/DZ = 150 µm, higher magnification = 40 µm.

**Figure 3 cells-12-01687-f003:**
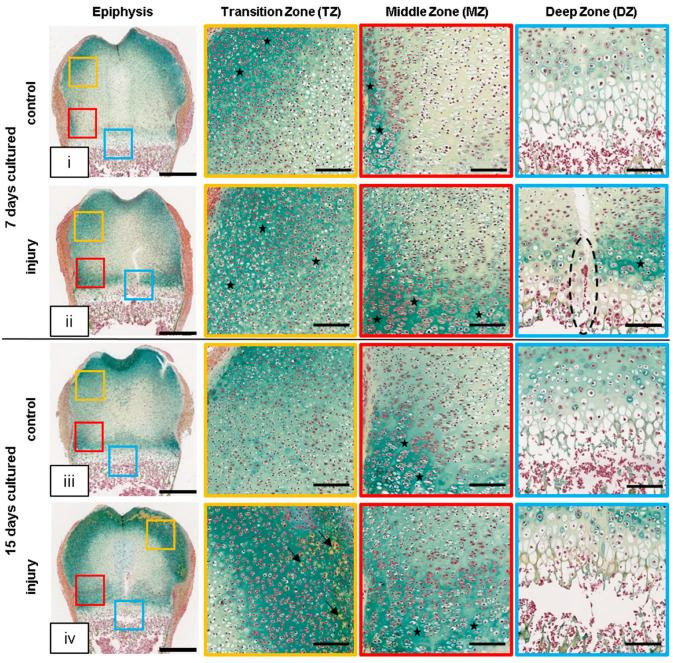
Injury induces tissue matrix mineralization. (**i**–**iv**) Representative images of OTC at day 7 and day 15 and their respective controls stained with Movat’s Pentachrome staining. Overview of rat distal femur with and without injury, colored squares represent the transition zone (TZ, yellow), middle zone (MZ, red), and deep zone (DZ, blue) at higher magnification. Note the cell aggregation within the DZ of injured bone slices at day 7 (black dashed oval). Cellular structures were stained as follows: cell nuclei (black), cytoplasm (reddish), elastic fibers (red), mineralized tissue (bright yellow, black arrow), and cartilage (green; black stars); scale bar: epiphysis = 500 µm, TZ/MZ/DZ = 150 µm.

**Figure 4 cells-12-01687-f004:**
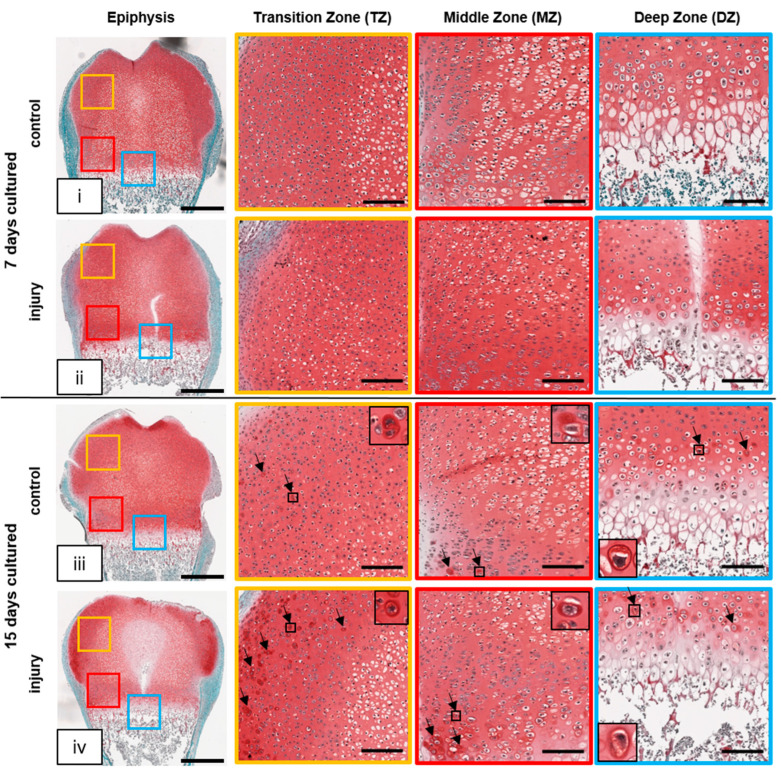
Proteoglycan (PG) distribution and aggregation are influenced by growth plate injury. (**i**–**iv**) Representative images of OTC at day 7 and day 15 and their respective controls stained with Safranin O/Fast Green to evaluate ECM composition changes. Overview of rat distal femur with and without injury, colored squares represent the transition zone (TZ, yellow), middle zone (MZ, yellow), and deep zone (DZ, blue). Cellular structures are stained as follows: cell nuclei (black), cytoplasm (green bluish), and cartilage—proteoglycans (PG; red). Black arrows = active chondrocytes producing PG. Scale bar: epiphysis = 500 µm, TZ/MZ/DZ = 150 µm, higher magnification = 40 µm.

**Figure 5 cells-12-01687-f005:**
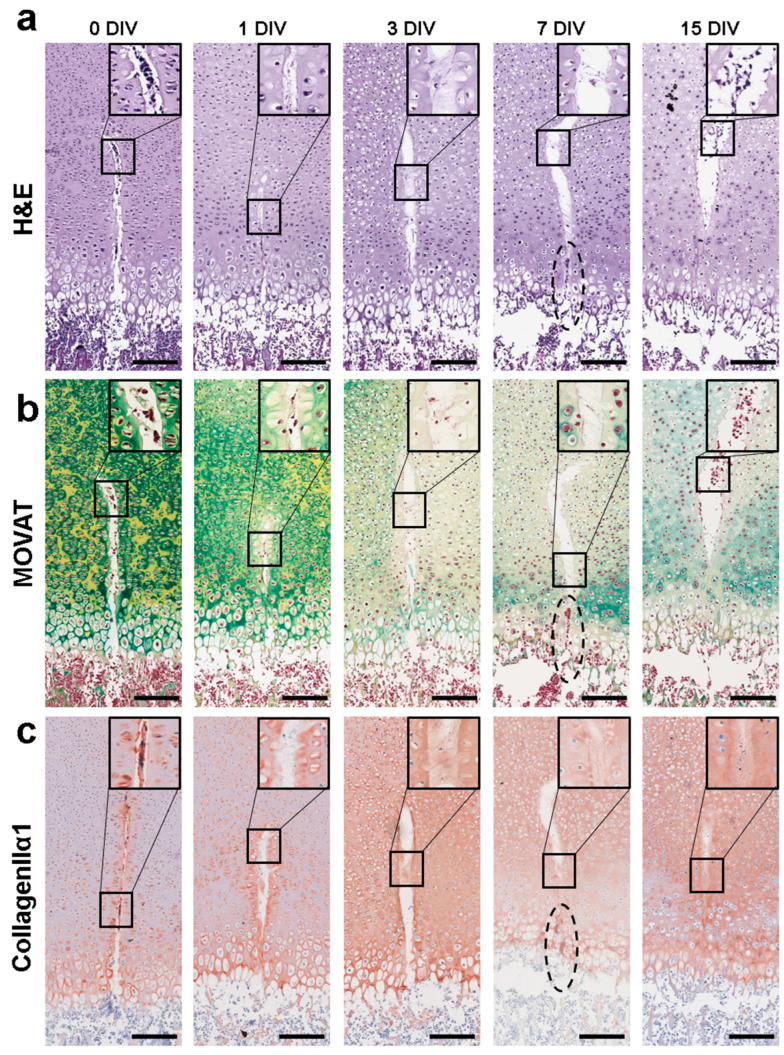
Growth plate injury regeneration of ex vivo bone slices under in vitro culture. To assess growth plate (GP) injury regeneration we performed (**a**) hematoxylin and eosin (H&E) and (**b**) Movat’s Pentachrome histological staining, and (**c**) type-IIα1 collagen (Col2α1) immunohistochemistry. Representative images of bone slices at 0, 1, 3, 7, and 15 days in vitro (DIV) are shown with higher magnification of newly formed structures within the injury wound. Black dashed oval = cell aggregation, which was detected within all performed stainings. Scale bar = 150 µm.

**Figure 6 cells-12-01687-f006:**
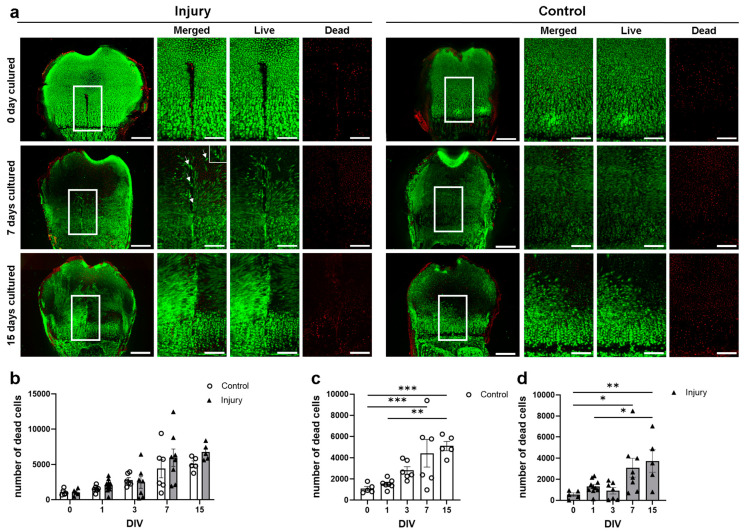
Determination of tissue viability surrounding the growth plate injury and survival in vitro. (**a**) Live and dead staining was performed to assess tissue viability by AM and EthD-1. Imaging of injured bone at 0, 7, and 15 days in vitro (DIV) was performed using a laser confocal microscope. Green = live cells, red = dead cells; white arrows—elongated cells; scale bar = 150 µm; higher magnification = 40 µm; (**b**–**d**) Dead cells within the injury area were quantified and the statistical significance of the observed difference was tested with Mann–Whitney U test (change by treatment) or Kruskal–Wallis test (temporal changes). Data are summarized as the mean ± SEM, and actual data were overlaid on histograms as triangles (injury) and circles (control) n ≥ 4, * *p* ≤ 0.05, ** *p* ≤ 0.01, *** *p* ≤ 0.001.

**Figure 7 cells-12-01687-f007:**
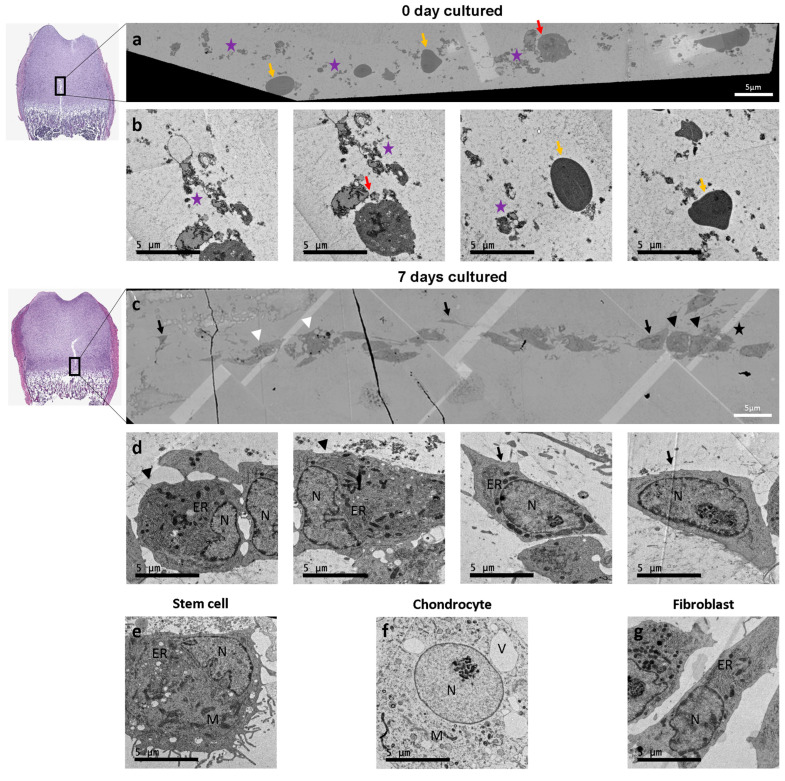
Scanning transmission electron microscopy (STEM) and conventional imaging of an ex vivo growth plate injury bone slice model. (**a**,**c**) STEM imaging at (**a**) day 0 and (**c**) day 7 of the in vitro culture providing an overview of a larger area of the injury site. (**b**,**d**) Conventional microscopy of the located area at (**b**) day 0, indicating the presence of cell fragments (purple stars), erythrocytes (yellow arrow), and chondrocytes (red arrow), and (**d**) at day 7, indicating stem cells (black arrowheads), chondrocytes (white arrowhead), fibroblasts (black arrow), and collagen fibers (black star). (**e**) Electron micrographs of stem cells have a fragmented nucleus (N), dilated endoplasmatic reticulum (ER), and mitochondria (M); (**f**) chondrocytes have a central nucleus (N), mitochondria (M) and vacuoles (V); and (**g**) fibroblasts have an elongated cell shape, nucleus (N) and endoplasmatic reticulum (ER). Scale bar = 5 µm.

**Table 1 cells-12-01687-t001:** To analyze ECM composition, aggrecan (Acan), type II collagen (Col2α1), and type X collagen (ColX) were used. Each antibody was optimized with respect to antigen retrieval, protein and hydrogen peroxidase blocking, and antibody dilution.

Primary Antibody	Company	Host	HIER	H_2_O_2_ Block	Protein Block	Dilution
Acan	Abcam (ab36861)	rabbit	EDTA	30 min	35 min	1:300
Col2α1	Invitrogen (MA5-12789)	mouse	Sodium citrate	25 min	25 min	1:100
ColX	Invitrogen (14-9771-82)	mouse	EDTA	10 min	25 min	1:100

## Data Availability

All available data are presented in this article and the corresponding Appendix A.

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
