# Peer review of "Disruption of Endochondral Ossification and Extracellular Matrix Maturation in an Ex Vivo Rat Femur Organotypic Slice Model Due to Growth Plate Injury"

_cells, 2023, doi:10.3390/cells12131687_

Round 1

Reviewer 1 Report

In this study, Etschmaier et al. develop a bone growth plate lesion model practiced on the sliced femur of Sprague Dawley rats that were cultured ex vivo. The histological study of the cultured bone slices recognizes a disruption of the bone growth plate as a consequence of the performed lesion. The induced situation affects both the cellular and the extracellular matrix components. The descriptive work carried out by the authors leads them to conclude that the developed model is an appropriate tool to study bone pathologies.

Comments:

It would be advisable to show a histological view of the lesion at time 0 in Figure 1. Although this information appears as Supplementary data, would be important showing a histological section that clearly shows the maximum extension of the lesion once it is practiced in the sample.

Histological work done by the authors is remarkable. To provide a higher quality of data, it would be interesting to quantitatively evaluate the cellular phenotypes that occur at the site of the lesion practiced. In this sense, it would be important to relativize this information taking into account the lesion area studied in each histological section.

In the discussion section, the authors acknowledge that one of the limitations of the organotypic femur culture model is the lack of vascularity. The authors rely on existing literature where it is suggested that this factor is not critical in short-term studies of bone development. This argument excludes the involvement of the immune system in the cartilage repair process. However, in many studies, the relationship between the immune system and bone repair is very important, including the early times of the process onset. In order to consider that the proposed model makes a relevant contribution to understanding established bone pathologies, it should be studied under conditions where the systemic influence plays a conditioning factor. Thus, it can be safely analyzed the limits of the application scope of the developed model. An example of this type of study could be those where the obtained bone slices are co-cultured with blood cells from syngeneic animals.

None

Author Response

In this study, Etschmaier et al. develop a bone growth plate lesion model practiced on the sliced femur of Sprague Dawley rats that were cultured ex vivo. The histological study of the cultured bone slices recognizes a disruption of the bone growth plate as a consequence of the performed lesion. The induced situation affects both the cellular and the extracellular matrix components. The descriptive work carried out by the authors leads them to conclude that the developed model is an appropriate tool to study bone pathologies.

It would be advisable to show a histological view of the lesion at time 0 in Figure 1. Although this information appears as Supplementary data, would be important showing a histological section that clearly shows the maximum extension of the lesion once it is practiced in the sample.

Comments: Thank you for your suggestion. We added an H&E overview staining at time 0 to the Fig.1c to demonstrate the maximum injury, which was implemented.

Histological work done by the authors is remarkable. To provide a higher quality of data, it would be interesting to quantitatively evaluate the cellular phenotypes that occur at the site of the lesion practiced. In this sense, it would be important to relativize this information taking into account the lesion area studied in each histological section.

Comments: Thank you for recognizing the quality of our histological work. This study describes an effort to establish an ex vivo model for growth plate injury, which, to the best of our knowledge, does not exist yet. To conduct this study, we prepared our organotypic femur culture model (OTC) from 4-day-old rat pups. Although all pups were the same age, they varied in weight (12.0 g +/- 2.1 g) and size. The injury was manually inflicted starting at the 2nd ossification center. Due to the methodology employed, we believe that a quantitative analysis of the lesion area is not meaningful. In our future studies, we aim to further modify our ex vivo culture model and incorporate the suggested additional quantitative assessments.

In the discussion section, the authors acknowledge that one of the limitations of the organotypic femur culture model is the lack of vascularity. The authors rely on existing literature where it is suggested that this factor is not critical in short-term studies of bone development. This argument excludes the involvement of the immune system in the cartilage repair process. However, in many studies, the relationship between the immune system and bone repair is very important, including the early times of the process onset. In order to consider that the proposed model makes a relevant contribution to understanding established bone pathologies, it should be studied under conditions where the systemic influence plays a conditioning factor. Thus, it can be safely analyzed the limits of the application scope of the developed model. An example of this type of study could be those where the obtained bone slices are co-cultured with blood cells from syngeneic animals.

Comments: We appreciate the thoughtful comment regarding the limitations of our organotypic femur culture model. In future investigations, we will consider conducting studies that incorporate systemic influences, such as co-culturing the bone slices with blood cells. These additional experiments will help further analyze the model's applicability and provide insights into its potential limitations. We appreciate your valuable input and will take it into consideration for our future research.

The manuscript was corrected again by a native speaker to avoid linguistic errors.

We appreciate your valuable feedback and will strive to incorporate your suggestions in our future research endeavors.

Reviewer 2 Report

This manuscript hypotheses that GP injury in this ex vivo organotypic femur model will lead to 69 pronounced alterations in the highly structured endochondral osteogenesis processes. The authors established an ex vivo culture of 3D culture of bone slices. They examined the tissue architecture and changes in cell population, including chondrocytes, mesenchymal stem cells (MSC), osteoblasts, and chondroclasts, during physiological and pathophysiological growth plate regeneration.

They discovered that injury-induced disruption of the highly structured endochondral ossification process towards an augmented cartilage generation and an attenuated mineralization/ossification process. The result may be different with the scenario in vivo due to a shortage of blood vessels.

The discovery supplied information to the community. More importantly, the bone slice culturing model provides a valuable model for ex vivo analysis of bone regeneration.

Author Response

Authors response: We would like to thank you very much for the positive assessment of our work and thank the reviewer for his efforts and time.

Reviewer 3 Report

The authors did a nice job to answer their scientific questions. However, there are some comments to improve the manuscript:

1.     In the “Introduction” section, most of the paragraphs are missing the references at the end.

3.     Line 81, what is the rationale for choosing day points 1, 3, 7, and 15? Please provide a reference.

4.  Line 86: What do the authors mean by “Depending on the downstream performed analysis”?

5.  The “experimental design” paragraph is the repetition of the following methodology. I suggest instead of this paragraph, provide an illustration image, please.

6.  The definition of control was first mentioned in line 133 as “respective controls”. I suggest explaining the control group in more detail and in the beginning of the methodology.

7. There are a lot of repetition phrases, please fix it. For example, lines 89-90 is repeated in lines 164-165. Line 229, the phrase growth plate (GP) has been mentioned before and GP term has been used, and again growth plate (GP) phrase is written in line 229. Line 402, repetition of days in vitro (DIV) which has been mentioned before. There are more examples, please fix this issue in the entire manuscript.

8.     Line 189, Please provide a reference for lines 189-190.

9.     Table 1, please provide the catalog number for each antibody.

10.  Line 196, the authors mentioned chondrogenesis and osteogenesis were assessed with immunohistochemistry. However, all the antibodies used in this study are for chondrogenesis.

11.  Line 216, please explain how many measurements were analyzed per replicate.

12.  Figure 4. iv, the red square is chosen from a site that looks like it has been broken. Would the authors please provide another representative image that does not have such an artifact?

13.  Line 354, how did the mineralization/ossification process was assessed?

14.  It seems the authors rely on the nucleus morphology to identify MSC and chondrocytes for STEM analysis. This information is written in the supplementary. Please add this information in the methodology section. Also, the reviewer didn’t find a description of how they identified the fibroblast in STEM analysis.

The E

Author Response

The authors did a nice job to answer their scientific questions. However, there are some comments to improve the manuscript:

  1. In the “Introduction” section, most of the paragraphs are missing the references at the end.

Authors response: We added the references at the end of the paragraphs. However, the last paragraphs contain our hypothesis and the reasoning for the study and therefor do not need references in our opinion.

  1. Line 81, what is the rationale for choosing day points 1, 3, 7, and 15? Please provide a reference.

Authors response: We included a reference and added the following sentence in line 122-126 “Slices were analyzed to investigate distinct phases of the regeneration processes namly in-flammatory phase (days 1-3), fibrogenic phase (days 3-7) and osteogenic & maturation phase (days 8+) [14]. Therefor slices were harvested on the day of preparation (0) and days 1, 3, 7, and 15 of in vitro cultivation (see Fig.1a).”  

  1.  Line 86: What do the authors mean by “Depending on the downstream performed analysis”?

Authors response: As you suggested with point 5, we have deleted the “experimental design” paragraph. In line 122-126 we now mention at the end of the paragraph “Preparation and culturing of ex vivo organotypic bone slice culture (OTCs)” that “Slices were analyzed to investigate distinct phases of the regeneration processes namely inflammatory phase (days 1-3), fibrogenic phase (days 3-7) and osteogenic & maturation phase (days 8+) [14]. Therefor slices were harvested on the day of preparation (0) and days 1, 3, 7, and 15 of in vitro cultivation (see Fig.1a).

  1.  The “experimental design” paragraph is the repetition of the following methodology. I suggest instead of this paragraph, provide an illustration image, please.

Authors response: As you suggested we have deleted this paragraph. The experimental setup illustration is provided as first result and we added a reference to the illustration in the line 126.

  1.  The definition of control was first mentioned in line 133 as “respective controls”. I suggest explaining the control group in more detail and in the beginning of the methodology.

Authors response: Within the section “Preparation and culturing of ex vivo organotypic bone slice culture (OTCs)” we mentioned that the left leg is the injury bone slice and the respective right leg of the same animal serves as respective control (lines 111-112).

  1. There are a lot of repetition phrases, please fix it. For example, lines 89-90 is repeated in lines 164-165. Line 229, the phrase growth plate (GP) has been mentioned before and GP term has been used, and again growth plate (GP) phrase is written in line 229. Line 402, repetition of days in vitro (DIV) which has been mentioned before. There are more examples, please fix this issue in the entire manuscript.

Authors response: Thank you for bringing this to our attention. We have implemented a uniform phrasing within the manuscript.

  1. Line 189, Please provide a reference for lines 189-190.

Authors response: Reference was added.

  1. Table 1, please provide the catalog number for each antibody.

Authors response: We added the catalogue number in the table.

  1. Line 196, the authors mentioned chondrogenesis and osteogenesis were assessed with immunohistochemistry. However, all the antibodies used in this study are for chondrogenesis.

Authors response: We agree and therefore we have deleted the assessment of osteogenesis in this line.

  1. Line 216, please explain how many measurements were analyzed per replicate.

Authors response: Per bone slice we performed each staining 3 times at different depths. These together with the other performed experiments were analysed. We have added this information in line 226-228.

  1. Figure 4. iv, the red square is chosen from a site that looks like it has been broken. Would the authors please provide another representative image that does not have such an artifact?

Authors response: We replaced the image with another respective image.

  1. Line 354, how did the mineralization/ossification process was assessed?

Authors response: Mineralization/ossification was assessed qualitatively with the Movat-Pentachrome staining.

  1. It seems the authors rely on the nucleus morphology to identify MSC and chondrocytes for STEM analysis. This information is written in the supplementary. Please add this information in the methodology section. Also, the reviewer didn’t find a description of how they identified the fibroblast in STEM analysis.

Authors response: We have added the images for stem cells, chondrocytes and fibroblasts in the STEM results. We also added the information “Cells were identified based on their distinct morphological appearance, namely their nucleus and shape, which was compared to cells gained from the cell culture or control slices.” in the material and methods section (lines 169-171).

The manuscript was corrected again by a native speaker to avoid linguistic errors.

We would like to thank the reviewer very much for his/her efforts and time and hope that we have addressed all points to his/her satisfaction.